# Acute Effect of Fixed vs. Self-Selected Rest Interval Between Sets on Physiological and Performance-Related Responses

**DOI:** 10.3390/jfmk9040200

**Published:** 2024-10-21

**Authors:** Diego A. Alonso-Aubin, Juan Hernández-Lougedo, Alberto Cavero-Haro, Ismael Martínez-Guardado

**Affiliations:** 1Strength Training and Neuromuscular Performance Research Group (STreNgthP), Faculty of Health Sciences—HM Hospitals, University Camilo José Cela, C/Castillo de Alarcón, 49, Villanueva de la Cañada, 28692 Madrid, Spain; diegoalexandre.alonso@ucjc.edu (D.A.A.-A.); jlougedo@ucjc.edu (J.H.-L.); 2Faculty of Health Sciences, Camilo José Cela University, C. Castillo de Alarcón, 49, Villafranca del Castillo, 28692 Madrid, Spain; alberto.cavero@alumno.ucjc.edu; 3LFE Research Group, Department of Health and Human Performance, Faculty of Physical Activity and Sport Science (INEF), Universidad Politécnica de Madrid, Calle de Martín Fierro, 7, 28040 Madrid, Spain

**Keywords:** rest time, loss velocity, mean propulsive velocity, back squat, near-infrared spectroscopy

## Abstract

**Background:** Although the comparison between self-managed rest and fixed rest periods in subjects experienced in lower-limb strength training has been investigated, the results remain unclear due to controversies among some studies. Therefore, the present study aimed to analyze the role of self-managed rest versus fixed rest in athletic performance, mean propulsive velocity, velocity loss, muscle oxygen saturation, and rest time in trained subjects; **Methods:** Thirteen subjects with a minimum of one year of training experience (age (years): 26.31 ± 3.84; height (cm): 175.46 ± 5.61; weight (kg): 79.24 ± 6.83) were randomly assigned to two groups (self-selected rest group [SR] = 7 and fixed rest group [FR] = 6). The subjects underwent a session for evaluation (one maximum repetition (1RM) estimation, familiarization, and data collection) and another day for a traditional strength training session for the back squat, consisting of five sets of four repetitions at 80% of 1RM. One group took a fixed 2 min break, while the other group managed their breaks autonomously (resuming when they felt ready to perform the next set at maximum velocity). Mean propulsive velocity (MPV) was monitored using a linear position transducer, and muscle oxygen saturation (SmO_2_) was measured with a near-infrared spectroscopy device; **Results:** Significant differences between the groups were found for the rest time between the first and second sets (SR 97.29 ± 23.70 seg vs. FR 120 ± 0.00 seg). However, no differences were found for MPV, velocity loss, or SmO_2_; **Conclusions:** Given the similarities in performance and physiological outcomes between fixed and self-selected rest conditions, both can be used equally depending on the preferences and training goals of coaches and athletes.

## 1. Introduction

In the last decade, strength training has been recognized as highly important both for health [1] and athletic performance [2]. A key aspect of strength training is programming, which is defined as the design and development of programs within the training phases to achieve the desired physical conditioning effects [3]. Implementing a strength training program requires consideration of numerous important variables, such as frequency, training density, volume, intensity, exercise selection, exercise order, sets, repetitions, rest duration, and the type and duration of recovery [4].

One of the most important variables for achieving improvements in response to strength training, and the focus of this study, is the rest interval between sets. Generally, for intermediate (6 months of experience) and advanced (2 years of experience) training, the American College of Sports Medicine recommends rest periods of at least 2–3 min for multi-joint exercises using heavy loads. However, for single-joint exercises, shorter rest periods of 1–2 min are sufficient between sets [5].

Despite its importance, the recovery process between sets during resistance training has been an often overlooked aspect of the training process. The physiological processes underlying the recovery between sets in resistance training are complex and multifaceted [6]. During strength training, the body undergoes significant physiological stress, which can lead to the depletion of muscle glycogen stores and the accumulation of metabolic byproducts, such as lactic acid and hydrogen ions, which can interfere with muscle contraction [7]. To effectively recover and prepare for the next set, the body must replenish these depleted resources and clear the accumulated metabolites [8]. Moreover, resistance training demands high levels of central nervous system (CNS) activation, especially for motor unit recruitment, and rest time must allow for the full recovery of the CNS, which is essential for continued high force production [9]. From the muscle and tissue repair perspective, micro-tears occur in muscle fibers during resistance training, stimulating muscle repair and growth. Thus, supporting tissue repair during recovery is vital for long-term strength gains [10]. Furthermore, neurotransmitters like acetylcholine can be depleted at the neuromuscular junction during intense training. Thus, ensuring sufficient recovery at the neuromuscular junction supports sustained muscle activation, crucial for maintaining muscle contraction [11].

The length of rest between sets in strength training is a key factor influencing neuromuscular adaptations, with shorter rest intervals generally enhancing muscular endurance, while longer rest periods are better suited for developing maximal strength and muscle hypertrophy [2]. Schoenfeld et al. [12] found that short rest intervals of 60 seconds (s) are not ideal for maximizing muscle strength compared to 3 minutes (min) rest periods. Willardson and Burkett [13] also suggest that as strength gains progress, increasing training volume becomes necessary, which may initially require longer rest intervals (4–5 min) until the individual adapts and can perform the same volume with shorter rest periods of 2–3 min. Similarly, de Salles et al. [6] confirmed that when training with loads ranged from 50% to 90% for multiple sets, 3–5 min of rest between sets were needed to sustain repetition performance and avoid a significant drop in training intensity. Additionally, other reviews indicate that trained individuals benefit from longer rest periods (2 min or more) to maximize strength gains, while shorter (60 s) or moderate (60–120 s) rest intervals are sufficient for untrained individuals to achieve strength improvements [14].

Recent research has been comparing a new modality of rest between sets called self-selected rest with fixed rest intervals. However, there is limited research on this methodology during strength training. For instance, De Salles et al. [6] demonstrated no differences in the number of repetitions or total training volume between self-paced rest and fixed 2 min rest intervals, for both upper- and lower-body exercises. However, the authors determined that self-paced rest could be a beneficial option as it involves less recovery time than fixed rest and may reduce the overall duration of the training session, making it a more time-efficient strategy. Nevertheless, only one study has evaluated self-paced rest in both the short and long term (over 8 weeks) during upper-body strength exercises, demonstrating that self-paced rest improved the total number of repetitions compared to a fixed 75 s rest interval. However, both methods were effective in promoting muscle strength gains without significant differences across all exercises. In contrast, the fixed 75 s rest interval was 37% more time-efficient [15].

On the other hand, velocity-based training (VBT) is an increasingly popular autoregulation method that dynamically adjusts training loads to enhance resistance training [16]. This method allows us to monitor the mean velocity and mean propulsive velocity of all participants during each training session. These variables enable tracking of the speed at which a load is moved, allowing for the control of fatigue levels and perceived effort during training [17]. The application of velocity losses between 10% and 20% during strength training can help promote neuromuscular adaptations and reduce fatigue, thereby improving the quality of work. Velocity zones with real-time feedback can be used to enhance athletic performance [18]. According to Sánchez-Medina et al. [19], when quantifying this intervention, we know that greater velocity loss corresponds to higher metabolic demand and, consequently, greater fatigue (particularly related to lactate and ammonia). Therefore, the velocity loss method, measured with a linear transducer, is valid for objectively quantifying neuromuscular fatigue during strength training. Additionally, as shown by González-Hernández et al. [20], mean velocities during strength training were lower for groups with 1 min rest intervals between sets compared to those with 3 or 5 min rest intervals.

Another recently introduced method is the measurement of muscle oxygen saturation using near-infrared spectroscopy (NIRS), a metric that may be related to fatigue [21], due to reports that muscle oxygen saturation (SmO_2_) shows a decreasing trend after a muscle strength protocol [22]. Thus, NIRS can measure muscle oxygen saturation, indicating the muscular performance developed and the fatigue accumulated during resistance training [23]. Moreover, in the study by Lin et al. [24], it was observed that this device allowed for the detection of aerobic changes at the muscular level during strength training by measuring the reoxygenation time throughout the intervention. Nevertheless, to our knowledge, there are no previous studies comparing the muscle oxygen saturation responses of applying self-suggested or fixed intervals in body exercises performance.

Therefore, the main objective of this study was to analyze the short-term effects of a self-selected rest versus a fixed rest between sets on the sports performance of muscular strength in experienced subjects. Our initial hypothesis was that self-selected rest would result in better performance compared to fixed rest, with a shorter session.

## 2. Materials and Methods

### 2.1. Participants

Thirteen active adult men aged 20–35 with advanced experience in resistance training [25] were randomly divided into two groups: (a) fixed rest group (FR) with a duration of 2 min and (b) self-selected rest group (SR) (Table 1). To participate in the study, the following inclusion criteria were established: (a) more than 1 year of experience in strength training; (b) no cardiovascular, respiratory, metabolic, neurological, or orthopedic diseases or disorders that could affect the performance of the test; (c) no consumption of drugs or medications; (d) no smoking; (e) no consumption of sports supplements or pharmacological products during the study or in the 6 months prior to the start of the study. In addition, subjects were asked to refrain from their regular training program during the study intervention period. The protocol was reviewed and approved by the Camilo José Cela University bioethics committee (11_24_CrossfitRM) following the guidelines of the Declaration of Helsinki, updated at the World Medical Association Assembly in Fortaleza in 2013 for research on human subjects. Each participant was assigned a code to maintain their anonymity.

### 2.2. Design

The present intervention proposal is based on a counterbalanced quasi-experimental design and groups were established at random using a specific software (https://www.randomizer.org (accessed on 1 May 2024)). The procedures applied in the study lasted one week, during which two visits were made to the laboratory (Figure 1), separated by at least 48 h to avoid the influence of fatigue.

The first visit served as a familiarization session. In the second session, the training protocol was performed under specific rest time conditions. The exercise performed in each session was the back squat. This was carried out with the participants wearing athletic shoes and clothing, ensuring the same total volume for all participants, but distributed differently across the loads used. The dependent variables analyzed were mean propulsive velocity, SmO_2_, total rest time and rating of perceived exertion (RPE).

### 2.3. Procedures

#### 2.3.1. Physiological Variables

Muscle Oxygen Saturation: To assess muscle oxygen saturation, an NIRS device (MOXY, Hutchinson, MO, USA) was placed on the vastus lateralis of the quadriceps [21]. This device calculates the relative concentration of oxyhemoglobin (HbO_2_) in relation to the total amount of hemoglobin. Using this device, oxygen saturation was analyzed during different stages of the exercise: (1) At the beginning of the exercise (representing the initial SmO_2_), (2) at the end of the exercise (final SmO_2_), (3) the percentage decrease in saturation, and (4) reoxygenation time. Once all the data were obtained, the average for all sets was calculated for each subject. Subsequently, the average for each group was calculated by training set. For placement on the vastus lateralis, the protocol established by Gómez-Carmona et al. [21] was followed, which includes the following steps: (1) Before placing the device, the placement area was shaved; (2) The device was placed in the center of the vastus lateralis, 15 cm from the upper pole of the patella; (3) The device was secured with a band (self-adhesive wrap) and then covered with a dark strap to prevent ambient light from directly affecting the device. The data obtained were stored via the Moxy PC application (Fortiori Design LLC, Minneapolis, MN, USA) and later analyzed using Golden Cheetah software in its version 3.6 (Golden Cheetah training software, https://www.goldencheetah.org/ (accessed on 10 June 2024)).

#### 2.3.2. Mechanical Variables

Movement Velocity: This allowed us to determine the differences in mean propulsive velocity (MPV) between the two groups. The device used in this study was Vitruve (v2.0, Speed4Lifts, Madrid, Spain) due to its validity and reliability as a simple measurement tool for assessing mean velocity and MPV [26] at which a given load is moved. Vitruve operates at a frequency of 1000 Hz, providing one data point per millisecond in real time, with an installation time of 2.5 min. This device was attached to the barbell before performing the exercise.

RM test back squat: The measurement of this variable was performed following the guidelines of Banyard et al. [27]. Prior to testing, subjects warmed up on a stationary bicycle for 5 min at 75 W. Afterwards, the initial external load for the load test was set at 20 kg and was progressively increased in 10 kg increments until MPV dropped below 0.50 m·s^−1^. From this point, the load was progressively increased by 5 kg to 1 kg until the subjects were able to perform a single maximum lift, which was determined as the one maximum repetition (1RM). Three attempts were made at each lighter load (MPV > 1.00 m·s^−1^), two attempts for moderate loads (0.65 m·s^−1^ ≤ MPV ≤ 1.00 m·s^−1^), and only one attempt for heavier loads (MPV < 0.65 m·s^−1^). Rest intervals between sets were set at 3 min, and the protocol was conducted similarly to the evaluation session proposed later. Two experienced observers were present on each side of the bar to ensure the safety and integrity of the participants during the tests.

Velocity loss: This variable was established to determine the percentage of performance loss in each of the series of the exercise protocol. In this sense, it was calculated as the percentage of loss in lifting velocity from the fastest repetition (usually the first) to the slowest (last) in each series. This variable was measured through the Vitruve (v2.0, Speed4Lifts, Madrid, Spain) linear position transducer.

#### 2.3.3. Perceptual Variables

Rating of Perceived Exertion (RPE): This was assessed using the OMNI-RES scale, a validated scale for measuring perceived exertion during resistance exercises for both the upper and lower body [28]. The questionnaire consists of ten items, with a value of 0 representing an extremely easy level of perceived exertion, and a value of 10 corresponding to an extremely hard level of difficulty. Perceived exertion was monitored after each training session.

### 2.4. Trainig Sessions

Before each session, participants performed a 5 min warm-up on the Echo Bike (ROGUE, Echo Bike, Columbus, OH, USA), with a perceived exertion level of 5–6 on the OMNI-RES scale. This was followed by a specific warm-up consisting of 10 squats with an empty bar, 5 squats with a load of 50% of the 1RM and 2 squats with a load of 80% of the 1RM for the back squat. After the warm-up, the training was carried out consisting of 5 sets × 4 repetitions at 80% of 1RM for the back squat exercise. This was the same for both groups and the only thing that differed was the rest time between sets. Both groups performed the training with the goal of performing each repetition at the maximum possible velocity in the concentric phase of the movement. To finish the training session, each subject cooled down by doing 5 min of Echo bike, with a perception of effort of 4–5 on the OMNI-RES scale.

### 2.5. Statistical Analysis

The data were processed using IBM SPSS 22.0 Statistics (IBM Corp., Armonk, NY, USA). A descriptive analysis was performed to show the means and standard deviations. The normality distribution of the variables was analyzed using the Shapiro-Wilk test and the homogeneity of variances using the Levene test. For parametric continuous variables in both groups that followed normal distributions and homogeneous variances, the Student T test for independent samples was used. For continuous variables that did not follow a normal distribution, the Mann–Whitney U test was performed. The statistical power was calculated at posteriori by effect size. The sample size (n = 13) was large enough to obtain an effect size value of 1.63, considering a sensitivity to detect real effects of change between 80 and 95%. The effect size (ES) of the intervention was calculated using Cohen’s guidelines [29]. Threshold values for ES were >0.2 (small), >0.6 (moderate), >1.2 (large) and >2.0 (very large). The percentage of change was determined. The *p* < 0.05 differences were considered statistically significant.

## 3. Results

The results obtained in this study are shown below. Table 1 shows the participant characteristics.

Table 2 shows mean velocity values for both rest configurations and Figure 2 shows individual mean velocity values across the sets. No significant differences were reported between the groups. However, a trend can be observed where the SR group progressively increases speed in each series, and the FR group decreases it as the training progresses.

Table 3 shows mean velocity loss values for both rest configurations and Figure 3 shows individual velocity loss percentage across the sets. No significant differences were reported between the groups. Nevertheless, it is observed that the velocity loss between sets 1 and 2 in the SR group is greater than in the FR group. The same pattern occurs between sets 3 and 4. Another important point to note from the table is that the fixed rest group shows a more stable and lower velocity loss during the first four sets, becoming equal to the SR group in the fifth set.

Table 4 shows mean SmO_2_ values for both rest configurations. Despite no differences between groups being observed, both groups increased muscle O_2_ saturation from the first to the fourth set, finally decreasing in the last set, in which we can observe a moderate effect size (0.77).

Figure 4 shows the rest time used by each group in each of the series. Only the first rest time showed significant differences between groups (*p* = 0.044), existing a difference of approximately 20 s between the SR vs. FR group. Another peculiarity that could be observed is that the DA group needed to increase the rest time in each series, with the time used by the SR in the last series being longer than that of the FR.

Figure 5 shows the RPE values for each group at the end of the training session. No significant differences were found for this variable, however the SR group reported lower values than the FR group (6.86 ± 1.07 and 7.17 ± 0.98 AU, respectively).

## 4. Discussion

The purpose of this study was to evaluate the short-term effects of a self-selected rest versus a fixed rest between sets, as well as on some performance and physiological parameters directly related to the sport.

Regarding the mean execution velocity, we can observe in the present study that this parameter was similar in both rest groups. However, the study by González-Hernández et al. [20] shows that subjects who rested for 3 or 5 min had a significantly higher mean velocity compared to those who rested for 1 min. Our hypothesis was that the self-managed rest group would yield a higher average velocity compared to the fixed rest group. However, what we observed was that the SR group exhibited an upward trend in average velocity from the first to the penultimate series, which coincided with the increase in rest time as the series progressed. This trend includes increases in average velocity in the third and fifth series, suggesting a conscious self-regulation by the subjects to perform the next series at the maximum propulsive velocity, avoiding fatigue. Therefore, it could be inferred that this trend, where the mean velocity increases as the series progress, might be due to the increase in rest time in the SR group as the series advance, aiming to reduce the neuromuscular fatigue induced by strength training. This mechanism could be attributed to metabolic and/or biochemical changes (increases in blood lactate, reduction in intramuscular pH levels, and acute hypoxia) rather than central fatigue [12]. In this line, this fact was already known thanks to Ibbott et al. [30], who explained in their study that those who took shorter self-managed rest periods exhibited lower power output in the squat. It was also observed that the fixed rest (FR) group experienced a stable average velocity until the third series and a marked decrease until the fifth series, which could coincide with neuromuscular fatigue associated with training.

According to velocity loss, it has been previously established that for strength training to remain effective, velocity loss should not exceed 20% (moderate velocity loss) [31]. This approach ensures an increase in strength, performance, and hypertrophy without adaptations that could negatively impact the neuromuscular performance of trained athletes. Another study, such as that by Galiano et al. [32], shows that for increasing maximum strength (1RM), a velocity loss of 5% induced the same gains as a 20% velocity loss, despite the fact that the subjects in the 5% velocity loss group performed only one-third of the repetitions compared to the other group. If we extrapolate this to our study, we observe that both the SR and FR groups experienced a maximum velocity loss of approximately 20% in the final series. This could provide valuable information, as it suggests that strength, performance, and hypertrophy could be increased through traditional strength training with corresponding self-managed rest periods. Additionally, other variables such as the countermovement jump (CMJ), 20 m sprint time, and average velocity for loads moved at both greater and less than 1 m/s could also be improved [32]. Other research supports the validity of using velocity loss to objectively quantify neuromuscular fatigue during strength training. This same study was able to establish that a maximum velocity loss of 30% for squatting could prevent an increase in blood ammonia above resting levels and decrease performance. When compared to our study, it could be hypothesized that by being below 20% of velocity loss in squatting, there would not be this accumulation of blood metabolites above resting levels, and therefore there would be no loss of performance.

In relation to SmO_2_, Gómez-Carmona et al. [21] established that pre-exercise oxygen saturation ranges between 77.30 ± 7.00% and 76.34 ± 5.44%, and post-exercise between 9.50 ± 9.70% and 7.30 ± 9.30%. Other studies, such as that by Davis et al. [33], also showed that the maximum values of muscle oxygen saturation were between 77.9 ± 1.5% and 68.6 ± 2.6%, while the minimum values ranged between 24.7 ± 2.3% and 22.2 ± 3.5%. This pattern was also observed in the study by Timon et al. [34], where the highest values ranged from 75.4 ± 9.5% to 73.6 ± 8.0%, and the lowest values from 20.4 ± 18.2% to 19.9 ± 16.8%. This is something we can observe in our study as well, where the highest oxygen saturation levels were 70.8% in the FR group and 83% in the SR group. Conversely, the lowest values were similar in both groups, 48%. However, in line with previous research [35], the minimum SmO_2_ values reported by our subjects would indicate that they experienced moderate fatigue, slightly decreasing performance, but not necessarily extreme fatigue, since there were still oxygen reserves to face the following sets. It is possible that this was due to the low number of repetitions carried out in each set, with the sets with a higher number of repetitions inducing a complete depletion of the muscle group required for the activity. Thus, this could confirm that, in strength training, lower SmO_2_ is usually associated with greater fatigue. When this variable decreases, the muscle depends more on the anaerobic pathway, generating a greater accumulation of waste products such as lactate [36]. To reduce these metabolites, rest time in strength training must be adjusted to allow for optimal reoxygenation between sets.

Another relevant characteristic observed in our study is the increase in oxygen saturation in both groups during the first and fourth sets, followed by a subsequent decrease in saturation in both groups during the fifth set. These results were also found in the study by Davis et al. [33], where saturation increased in the second set and decreased in the final set. This could be due to the warm-up not inducing sufficient metabolic, neuromuscular, and temperature adaptations. As a result, these adaptations may have been acquired later during the training session, potentially compromising the first part of the workout.

Regarding rest time, De Salles et al. [37] showed that self-managed rest periods for the squat were shorter during the first and second rest intervals compared to fixed rest, leading to no differences in the number of repetitions. The same occurred in the present study, resulting in a reduction in the training time for the SR group compared to the FR group. In another study, Schoenfeld et al. [38] compared a 1 min rest group to a 3 min rest group. This study demonstrated that the group with longer rest intervals produced greater increases in muscle strength (less velocity loss) compared to the group with shorter rest intervals. This could be counterproductive for the self-managed rest group, as this study observed that our subjects, on average, rested less than 2 min across all sets.

This suggests that it could be a disadvantage in terms of increasing muscle strength. However, Simão et al. [15] demonstrated that the self-managed rest group increased the number of repetitions compared to the fixed rest group. It is important to note that in that study, the fixed rest was 75 s, and the exercises involved the upper body (chest press, lat pull-down, shoulder press, and seated row). Therefore, caution is needed when comparing these results. Other findings from this study were that the fixed rest group (75 s) was 37% more efficient than the self-managed rest group. However, other researchers, such as Goessler et al. [39] observed no significant differences in the maximum number of repetitions between the SR vs. FR (2 min), but there were differences between the two fixed rest groups (2 min vs. 1 min) for the squat exercise. Both these studies and ours clarify that there are no differences between fixed rest and self-managed rest. In these studies, the participants in the SR group averaged around 2 min, without reaching 3 min in most cases. Therefore, given the lack of differences in the results of the compared studies, we could suggest that this strategy could save time in strength training for trained individuals when compared with other studies that propose 3–5 min rest periods to increase absolute strength in terms of chronic adaptations [6].

The present study is not free of limitations: (i) the sample size; (ii) the study only considers a single training session in an acute manner, so it would be interesting to conduct a long-term protocol; (iii) only men participated; (iv) no subjective psychological variables were measured that would explain the rest pattern of the group that rested independently. However, further research is needed to overcome the limitations found, as well as to obtain more consistent results.

## 5. Conclusions

Considering the results obtained, both self-selected rest and fixed rest have comparable effects on the variables evaluated, including average velocity, average velocity loss, average muscle oxygen saturation, and rest time. These findings suggest that the choice between self-managed rest and fixed rest can be based on individual preferences or training planning without significantly compromising athletic performance. Therefore, it can be concluded that both rest methods are equally valid for optimizing performance in individuals with strength training experience.

## 6. Practical Applications

As a practical application, self-selected rest could be considered a valuable tool for strength training, as it could offer the advantage of shorter training sessions to optimize time between sessions, despite no significant differences compared to fixed rest in the present study. This is an interesting point, given that effectiveness and efficiency are increasingly sought after in strength training for any type of population today.

## Figures and Tables

**Figure 1 jfmk-09-00200-f001:**
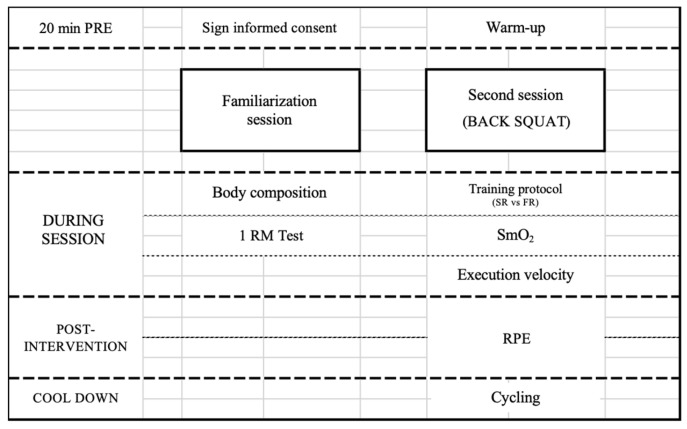
Schedule of assessments performed in the study.

**Figure 2 jfmk-09-00200-f002:**
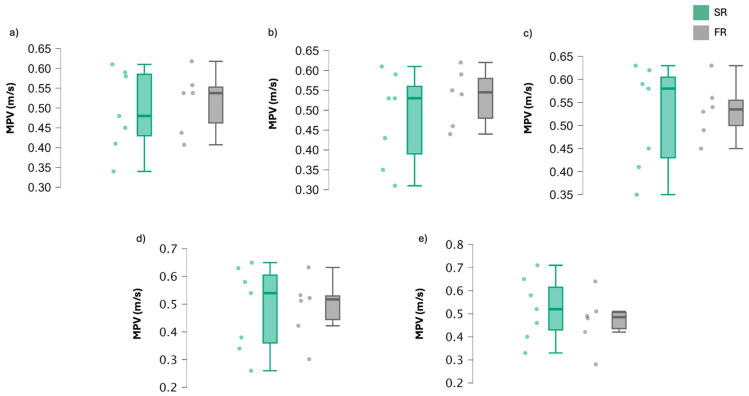
Values of mean propulsive velocity per subject and group in the five sets of back squat exercise. SR: self-selected rest group; FR: fixed rest group. (**a**): first set; (**b**): second set; (**c**): third set; (**d**): fourth set; (**e**): fifth set.

**Figure 3 jfmk-09-00200-f003:**
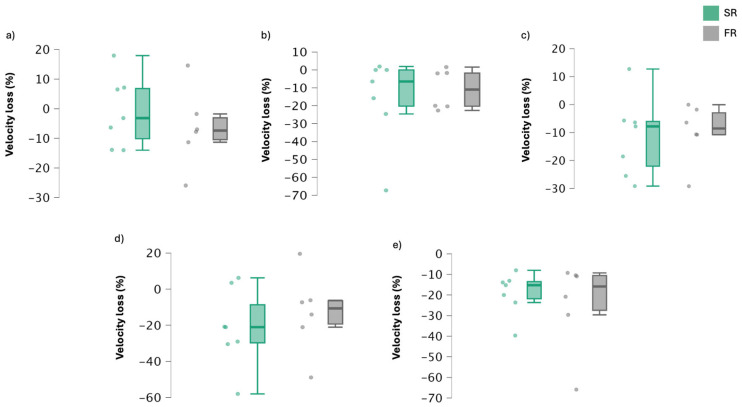
Values of velocity loss per subject and group in the five sets of back squat exercise. SR: self-selected rest group; FR: fixed rest group. (**a**): first set; (**b**): second set; (**c**): third set; (**d**): fourth set; (**e**): fifth set.

**Figure 4 jfmk-09-00200-f004:**
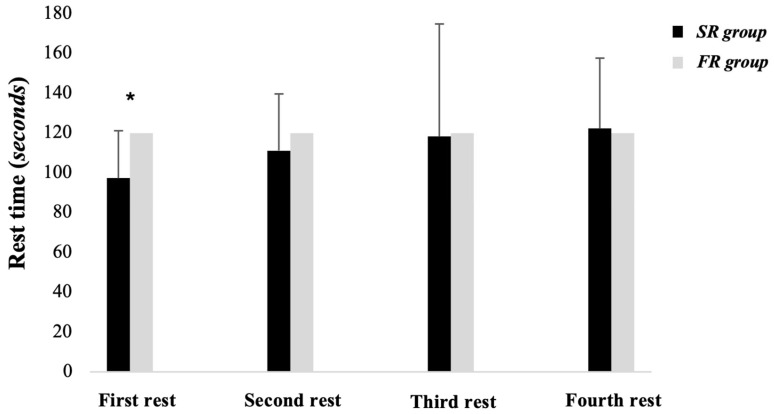
Rest time used for both groups during all sets. * Differences between groups.

**Figure 5 jfmk-09-00200-f005:**
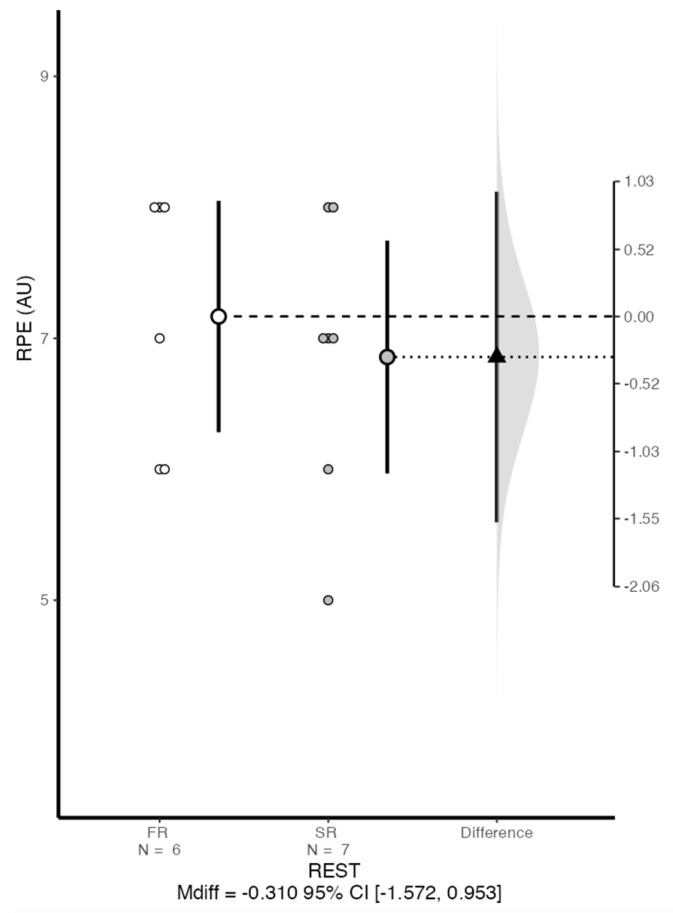
Individual values and mean of RPE with the difference between groups at the end of the session. SR: self-selected rest group; FR: fixed rest group.

**Table 1 jfmk-09-00200-t001:** Participant and training characteristics.

Parameters	SR (Mean ± Sd)	FR (Mean ± Sd)
Age (years)	26.71 ± 3.94	25.67 ± 4.22
Height (m)	1.75 ± 5.12	1.75 ± 6.62
Weight (kg)	81.84 ± 5.77	76.20 ± 7.16
Body mass Index	26.77 ± 1.30	24.92 ± 2.49
Lean body mass (kg)	65.56 ± 5.12	63.23 ± 3.18
Fat mass (%)	15.69 ± 4.16	13.08 ± 5.63
Bone Mass (kg)	3.40 ± 0.24	3.36 ± 0.28

**Table 2 jfmk-09-00200-t002:** Mean velocity values during all sets of back squat exercise for both groups.

		Set 1	*p*	*d*	Set 2	*p*	*d*	Set 3	*p*	*d*	Set 4	*p*	*d*	Set 5	*p*	*d*
MPV (m/s)	SR	0.49 ± 0.11	0.64	−0.32	0.49 ± 0.13	0.34	−0.38	0.51 ± 0.12	0.37	−0.21	0.52 ± 0.16	0.28	0.29	0.52 ± 0.14	0.26	0.40
	FR	0.52 ± 0.08			0.53 ± 0.08			0.53 ± 0.07			0.48 ± 0.12			0.47 ± 0.11		

SR: self-selected rest group; FR: fixed rest group; *p*: *p*-value between groups; d: effect size.

**Table 3 jfmk-09-00200-t003:** Mean velocity loss during all sets of back squat exercise for both groups.

		Set 1	*p*	*d*	Set 2	*p*	*d*	Set 3	*p*	*d*	Set 4	*p*	*d*	Set 5	*p*	*d*
Velocity loss (%)	SR	−0.83 ± 11.90	0.42	0.46	−16.03 ± 24.59	0.64	−0.29	−11.49 ± 14.25	0.81	−0.14	−21.37 ± 21.85	0.50	−0.38	−19.08 ± 10.38	0.56	0.04
	FR	−6.57 ± 13.21			−10.83 ± 11.24			−9.76 ± 10.45			−12.97 ± 22.33			−19.57 ± 11.40		

SR: self-selected rest group; FR: fixed rest group; *p*: *p*-value between groups; d: effect size.

**Table 4 jfmk-09-00200-t004:** Mean SmO_2_ values during all sets of back squat exercise for both groups.

		Set 1	*p*	*d*	Set 2	*p*	*d*	Set 3	*p*	*d*	Set 4	*p*	*d*	Set 5	*p*	*d*
SmO_2_ (%)	SR	64.87 ± 13.67	0.44	0.35	66.30 ± 13.77	0.40	0.35	66.12 ± 16.62	0.79	0.16	68.57 ± 13.82	0.72	0.37	62.20 ± 13.53	0.32	0.77
	FR	61.64 ± 4.81			62.98 ± 6.21			64.31 ± 5.75			64.88 ± 5.89			54.74 ± 5.93		

SR: self-selected rest group; FR: fixed rest group; *p*: *p*-value between groups; d: effect size.

## Data Availability

The data presented in this study are available on request from the corresponding author.

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
