# Peer review of "Acute Effect of Fixed vs. Self-Selected Rest Interval Between Sets on Physiological and Performance-Related Responses"

_jfmk, 2024, doi:10.3390/jfmk9040200_

Round 1

Reviewer 1 Report

Comments and Suggestions for Authors

First of all, the reviewer would like to thank the authors for their work and efforts in trying to improve sports science knowledge.

The authors aimed to analyze the role of selfmanaged rest versus fixed rest in Athletic performance, mean propulsive velocity, velocity loss, muscle oxygen saturation, and rest time in trained subjects..

Abstact:

Conclusions should be written in a hypothesis-driven manner. That is, the group with the self-administered rest does not produce gains over the other group, but this is also true the other way around. Conclusions should be written from a more neutral perspective.

Method;

- In the exclusion criteria, none of the exclusion criteria are stated about sports practice. That is, were the participants asked not to perform strength exercise in the last hour or days?

- The authors should justify why this sample size is adequate. Did they make an adequate sample size calculation? This is a very important question to solve.

Discussion:

In the introduction and discussion I miss that they address what aspects may affect the duration time of self-administered rest ( experience, pain perception, personality). Could it be affected by the situation of exposure to the study? Could it be affected by participants resting less because of a pressure to “recover soon” and a self-demand derived from the pressure of the study? It may be necessary to point out how these possible contaminating variables were addressed or how to address them in future research.

Author Response

First of all, the reviewer would like to thank the authors for their work and efforts in trying to improve sports science knowledge.

The authors aimed to analyze the role of selfmanaged rest versus fixed rest in Athletic performance, mean propulsive velocity, velocity loss, muscle oxygen saturation, and rest time in trained subjects..

Thank you very much for your time and for reviewing the manuscript, as your comments have helped us to clarify important aspects of the work that were not entirely clear before. We hope that you will like the proposed changes. 

Also, we apologize that the method describes the RPE, which is also measured, but we did not include the data in the results. You can now find them in this new version.

Abstact:

Conclusions should be written in a hypothesis-driven manner. That is, the group with the self-administered rest does not produce gains over the other group, but this is also true the other way around. Conclusions should be written from a more neutral perspective.

Thank you for your comment. This has been modified in the abstract:

Given the similarities in performance and physiological outcomes between fixed and self-selected rest conditions, both can be used equally depending on the preferences and training goals of coaches and athletes.

Method:

- In the exclusion criteria, none of the exclusion criteria are stated about sports practice. That is, were the participants asked not to perform strength exercise in the last hour or days?

Thank you for your comment. This has been specified in the document:

  • In addition, subjects were asked to refrain from their regular training program during the study intervention period was not allowed.

- The authors should justify why this sample size is adequate. Did they make an adequate sample size calculation? This is a very important question to solve.

Thank you for the comment. We also think that sample size is always important when conducting research. In our case, we recruited 13 subjects, which is in line with other studies that have evaluated the same independent variable (Do Carmo et al., 2021- DOI: 10.1519/JSC.0000000000002519; Goessler et al., 2013- https://doi.org/10.5007/1980-0037.2013v15n4p467; and Simao et al., 2022- DOI: 10.1519/JSC.0000000000003606). However, we have performed the calculation of the statistical power a posteriori through the effect size indicating the following in the manuscript:

  • The statistical power was calculated at posteriori by effect size. The sample size (n=13) was large enough to obtain an effect size value of 1.63, considering a sensitivity to detect real effects of change between 80-95%.

Discussion:

In the introduction and discussion I miss that they address what aspects may affect the duration time of self-administered rest ( experience, pain perception, personality). Could it be affected by the situation of exposure to the study? Could it be affected by participants resting less because of a pressure to “recover soon” and a self-demand derived from the pressure of the study? It may be necessary to point out how these possible contaminating variables were addressed or how to address them in future research.

  • It is true that training experience can significantly influence recovery between sets in strength training. Furthermore, for individuals with previous experience, self-suggested rest intervals can be an effective option and reduce total training session duration in strength training. In our study, we did not measure these more subjective variables that you mention, so we have included them in the limitations.

Reviewer 2 Report

Comments and Suggestions for Authors

Thank you for inviting me to review this manuscript for your journal. The study by Alonso-Aubin et al. attempts to validate a strength training planning methodology that consists of resting between sets as much as the athlete deems necessary. However, the study design and sample selection preclude making conjectures from their results. In general, the study is well written, but presents methological concerns that should be resolved.

1. I do not understand the connection of ideas that the authors are trying to establish in the third paragraph of the introduction (lines 49-61). I believe the concepts of training volume and intensity are being confused with those of density and performance. Please review the entire paragraph and clarify your message.

2. A brief introduction to the physiological processes that occur during recovery seems mandatory. In fact, the entire introduction provides a very superficial overview of the evidence and physiology underlying the recovery processes between sets.

3. Perhaps if the rationale of the study were clearer, it would help to focus the study’s hypothesis and, above all, give it a meaning that contributes to current knowledge. What is the actual hypothesis of the study? What is its contribution to existing knowledge?

4. In view of the results, only the rest after the first set was different between the groups, and there is a tendency to require more rest as the sets progress. The authors discuss that it would mean a more time-efficient training. How much time would you save? isn´t it negligible?

5. Although it may seem trivial, if velocity loss is to be used as a variable, it must be defined in the materials and methods.

6.  the variability found in the % of velocity loss prevents its use as a variable in this study, given the small sample size. 

7. It is known that the 1RM varies greatly from day to day, so establishing the load to be lifted by the participants based on the percentage of 1RM seems contradictory within the study itself. The mean propulsive velocity was also recorded so the actual load could have been established based on this.

8. In Tables 2, 3, and 4, the p-value is not shown.

9. The results show the average values of 6 or 7 young subjects with unequal experience in strength training, and the deviations between subjects are high, making it risky to draw ideas or conclusions from these data. In this regard, it would be useful to include tables with the results of each subject to see if the trends are the same for all of them or if there were better and worse responders or outliers.

10. the study does not present clear findings, and its results seem hardly generalizable to any population. I suggest that the authors delve deeper into the physiological mechanisms of fatigue and recovery, the different types or expressions of fatigue produced by resistance training, and the different physiological profiles that athletes may present.

Comments on the Quality of English Language

It is fine

Author Response

Thank you for inviting me to review this manuscript for your journal. The study by Alonso-Aubin et al. attempts to validate a strength training planning methodology that consists of resting between sets as much as the athlete deems necessary. However, the study design and sample selection preclude making conjectures from their results. In general, the study is well written, but presents methological concerns that should be resolved.

Thank you very much for your time and for reviewing the manuscript, as your comments have helped us to specify and include important aspects of the work that were not entirely clear before. We hope that you will like the proposed changes. 

Also, we apologize that the method describes the RPE, which is also measured, but we did not include the data in the results. You can now find them in this new version.

  1. I do not understand the connection of ideas that the authors are trying to establish in the third paragraph of the introduction (lines 49-61). I believe the concepts of training volume and intensity are being confused with those of density and performance. Please review the entire paragraph and clarify your message.
  • Thank you very much for your comment. It is true that the way the paragraph was composed could give rise to a misinterpretation, so we have restructured it for better understanding:

The length of rest between sets in strength training is a key factor influencing neuromuscular adaptations, with shorter rest intervals generally enhancing muscular endurance, while longer rest periods are better suited for developing maximal strength and muscle hypertrophy [2]. Schoenfeld et al. [6] found that short rest intervals of 60 seconds are not ideal for maximizing muscle strength compared to 3-minute rest periods. Willardson and Burkett [7], also suggest that as strength gains progress, increasing training volume becomes necessary, which may initially require longer rest intervals (4-5 minutes) until the individual adapts and can perform the same volume with shorter rest periods of 2-3 minutes. Similarly, de Salles et al. [8], confirmed that when training with loads ranging from 50% to 90% for multiple sets, 3-5 minutes of rest between sets are needed to sustain repetition performance and avoid a significant drop in training intensity. Additionally, other reviews indicate that trained individuals benefit from longer rest periods (2 minutes or more) to maximize strength gains, while shorter (60 seconds) or moderate (60-120 seconds) rest intervals are sufficient for untrained individuals to achieve strength improvements [9].

2. A brief introduction to the physiological processes that occur during recovery seems mandatory.In fact, the entire introduction provides a very superficial overview of the evidence and physiology underlying the recovery processes between sets.

  • Thank you for your comment, we agree with you that we were missing a physiological view of recovery between sets. Although it contains a multifaceted view, we have included a paragraph in the introduction indicating the main physiological events that can occur during strength training and how recovery can help restore them.

3. Perhaps if the rationale of the study were clearer, it would help to focus the study’s hypothesis and, above all, give it a meaning that contributes to current knowledge. What is the actual hypothesis of the study? What is its contribution to existing knowledge?

  • Thank you for your comment, we have added our initial hypothesis after the objectives for better understanding.

Therefore, the main objective of this study will be to analyze the short-term effects of a self-selected rest versus a fixed rest between sets on the sports performance of muscular strength in experienced subjects. Our initial hypothesis was that self-selected rest would result in better performance to the fixed rest, with a shorter session.

4. In view of the results, only the rest after the first set was different between the groups, and there is a tendency to require more rest as the sets progress. The authors discuss that it would mean a more time-efficient training. How much time would you save? isn´t it negligible?

  • Although our results are based on only one exercise with 5 sets, it is possible that in the development of a complete training session the self-managed rest strategy will be more efficient in terms of time compared to a fixed rest, as other authors have reported.

For better adaptation to our results, this question has been specified in the practical applications section:

As a practical application, self-selected rest could be considered a valuable tool for strength training, as it could be offers the advantage of shorter training sessions to optimize time between sessions, despite no significant differences compared to fixed rest in the present study. This is an interesting point, given that effectiveness and efficiency are increasingly sought after in strength training for any type of population today.

5. Although it may seem trivial, if velocity loss is to be used as a variable, it must be defined in the materials and methods.

  • Thank you very much for your comment. This has been added to methods section:

Velocity loss: This variable was established to determine the percentage of performance loss in each of the series of the exercise protocol. In this sense, it was calculated as the percentage of loss in lifting velocity from the fastest repetition (usually the first) to the slowest (last) in each series. This variable was measured through the Vitruve (v2.0, Speed4Lifts, Madrid, Spain) linear position transducer

6. The variability found in the % of velocity loss prevents its use as a variable in this study, given the small sample size. 

  • We agree with you, but we thought it might be interesting to observe the loss of performance over the sets with one rest or another. For better visualization, we have added a figure (Figure 3) where you can see the loss of mean propulsive velocity per subject and group over each of the sets. We hope you find it interesting.

7. It is known that the 1RM varies greatly from day to day, so establishing the load to be lifted by the participants based on the percentage of 1RM seems contradictory within the study itself. The mean propulsive velocity was also recorded so the actual load could have been established based on this.

  • We agree with you that the 1RM varies daily, so we estimate it through a linear position transducer as established in the method section corresponding to this variable. In addition, we have added that the protocol we based this on was Banyard et al. 2017 (DOI: 10.1519/JSC.0000000000001657).

8. In Tables 2, 3, and 4, the p-value is not shown.

  • Thank you very much for your comment. This has been added to tables.

9. The results show the average values of 6 or 7 young subjects with unequal experience in strength training, and the deviations between subjects are high, making it risky to draw ideas or conclusions from these data. In this regard, it would be useful to include tables with the results of each subject to see if the trends are the same for all of them or if there were better and worse responders or outliers.

  • Thank you for your contribution. Just as we have done with the velocity loss variable, we have made the same figure (Figure 2) for the data on mean propulsive velocity in which you can see the values ​​reached by each subject in each of the groups and set.

10. The study does not present clear findings, and its results seem hardly generalizable to any population. I suggest that the authors delve deeper into the physiological mechanisms of fatigue and recovery, the different types or expressions of fatigue produced by resistance training, and the different physiological profiles that athletes may present.

  • Thank you for your comment, we have included the physiological point of view in the introduction. We carried out the study because most of the previous studies only quantified the number of repetitions per group to determine which one had the best performance, self-selected or fixed rest. However, we believe that the measurement of the mean propulsive velocity, the loss of velocity between series and muscle oxygen saturation could indicate in a more objective way if the performance was different between groups in function to the rest type selected.

Reviewer 3 Report

Comments and Suggestions for Authors

Line 44-45

Generally, for intermediate (6 months of experience) and advanced (2 years of experience) training, …

Comment 1. Explain what division this - intermediate or advanced

Line 66-69

However, self-paced rest could be a beneficial option as it involves less recovery time than fixed rest and may reduce the overall duration of the training session, making it a more time-efficient strategy.

Comment 2. The claim that self-paced rest involves less recovery time should be supported by a reference.

Line 91-92 Another recently introduced method is the measurement of muscle oxygen saturation using near-infrared spectroscopy (NIRS), a metric that may be related to fatigue.

Comment 3. may be related - Please explain in more detail

Sample - more than 1 year of training experience

Comment 4. In most sports, they are beginners, in the aim it says that they are experienced subjects??

Comment 5. We do not know whether the respondents are former or active athletes or have never played sports.

Table 1

Comment 6. These data should be presented separately by group.

Line 309-325

SmO2 – chapter

Comment 7. It is not at all clear why saturation was measured in this study?

Final comment:

The sample is small and undefined, so let us find out who the results of this study may apply to. It is not clear why oxygen saturation was measured and the performance assessment tests rely solely on performance speed, which is not sufficient to draw high quality conclusions.

Author Response

Thank you very much for your time and for reviewing the manuscript, as your comments have helped us to specify and include important aspects of the work that were not entirely clear before. We hope that you will like the proposed changes.

Also, we apologize that the method describes the RPE, which is also measured, but we did not include the data in the results. You can now find them in this new version.

Line 44-45

Generally, for intermediate (6 months of experience) and advanced (2 years of experience) training, …

Comment 1. Explain what division this - intermediate or advanced

  • Thank you for your comment, this has been added to the participant’s section.

Line 66-69

However, self-paced rest could be a beneficial option as it involves less recovery time than fixed rest and may reduce the overall duration of the training session, making it a more time-efficient strategy.

Comment 2. The claim that self-paced rest involves less recovery time should be supported by a reference.

  • Thank you for your appreciation. It has been clarified in the document that this statement was established by the authors De Salles et al. 2016.

Line 91-92 Another recently introduced method is the measurement of muscle oxygen saturation using near-infrared spectroscopy (NIRS), a metric that may be related to fatigue.

Comment 3may be related - Please explain in more detail

  • Thank you for your comment. This has been specified in the document:

Another recently introduced method is the measurement of muscle oxygen saturation using near-infrared spectroscopy (NIRS), a metric that may be related to fatigue [16], due to has been reported that muscle oxygen saturation (SmO2) shows a decreasing trend after a muscle strength protocol [17]. Thus, NIRS can measure muscle oxygen saturation, indicating the muscular performance developed and the fatigue accumulated during resistance training [18].

Sample - more than 1 year of training experience

Comment 4. In most sports, they are beginners, in the aim it says that they are experienced subjects??

  • Thank you for your comment. In this sense and according to the training status classification tables carried out by Junior et al. 2021, our athletes would be classified as advanced in strength training by having more than one year of experience.
  • The DOI of the reference in question is the following. DOI: 10.1519/SSC.0000000000000627

Comment 5. We do not know whether the respondents are former or active athletes or have never played sports.

  • Thank you for your comment, this has been added to the participant’s section. In this line, they were active young men.

Table 1

Comment 6. These data should be presented separately by group.

  • Thank you for your comment. The table has been modified.

Line 309-325

SmO2 – chapter

Comment 7. It is not at all clear why saturation was measured in this study?

  • Thank you for your comment. As mentioned in the introduction, the measurement of muscle oxygen saturation can be an interesting parameter to detect fatigue during strength training, as it is a simple and non-invasive technique to apply. Therefore, we introduced it in our study to find out which type of rest produced greater muscle desaturation for the same exercise. This type of variable has not been previously introduced by any of the studies that have evaluated a fixed type of rest with another self-suggested one, so we thought that its inclusion could be interesting and that it could give us additional information on the recovery after series of both training groups.

Final comment:

The sample is small and undefined, so let us find out who the results of this study may apply to. It is not clear why oxygen saturation was measured and the performance assessment tests rely solely on performance speed, which is not sufficient to draw high quality conclusions.

  • Thank you for your comment, we have modified the questions you asked to clarify the characteristics of the sample. In relation to your other question, we introduced the measurement of the average propulsive velocity, the loss of velocity between series, because most of the previous studies only quantified the number of repetitions per group to determine which one had the best performance. However, we believe that the measurement of these variables in our study could indicate in a more objective way if the performance was different between groups.

Round 2

Reviewer 1 Report

Comments and Suggestions for Authors

Thank you for the effort made to improve the quality of the manuscript.

Author Response

Thank you for the effort made to improve the quality of the manuscript.

- Thank you very much for your time in the review process, your comments helped us improve the manuscript.

Reviewer 2 Report

Comments and Suggestions for Authors

I would like to thank the authors for the revisions made to their manuscript. The changes have significantly enhanced the clarity and quality of the work. I appreciate your responsiveness to the feedback provided, which has resulted in a more robust and coherent document.

I am pleased to note that the revised manuscript now addresses all the raised concerns and follows the suggestions given. Therefore, I consider it suitable for further processing.

Author Response

I would like to thank the authors for the revisions made to their manuscript. The changes have significantly enhanced the clarity and quality of the work. I appreciate your responsiveness to the feedback provided, which has resulted in a more robust and coherent document.

I am pleased to note that the revised manuscript now addresses all the raised concerns and follows the suggestions given. Therefore, I consider it suitable for further processing.

- Thank you very much for your time in the review process, your comments helped us improve the manuscript.

Reviewer 3 Report

Comments and Suggestions for Authors

The paper has been significantly improved, but the very brief part about fatigue measured with the MOKSI device is still under discussion. Are they tired? How tired are they? We know that there are no differences between the groups and that others have reported similar results. The question is what we should do with this knowledge now? Briefly complete this section.

Author Response

The paper has been significantly improved, but the very brief part about fatigue measured with the MOKSI device is still under discussion. Are they tired? How tired are they? We know that there are no differences between the groups and that others have reported similar results. The question is what we should do with this knowledge now? Briefly complete this section.

- Thank you very much for your time in the review process, your comments helped us improve the manuscript.

- We certainly also felt that we needed to establish what the oxygen saturation obtained by our subjects implies in our study. Therefore, we have added the following to that section and we hope you find the reflection made correct.

This is something we can observe in our study as well, where the highest oxygen saturation levels were 70.8% in the FR group and 83% in the SR group. Additionally, the lowest values were similar in both groups, 48%. However, in line with previous research [35], the minimum SmO2 values ​​reported by our subjects would indicate that they experienced moderate fatigue, slightly decreasing performance, but not necessarily extreme fatigue, since there are still oxygen reserves to face the following sets. It is possible that this was due to the low number of repetitions carried out in each set, being the sets with a higher number of repetitions inducing a complete depletion of the muscle group required for the activity. Thus, this could confirm that, in strength training, lower SmO2 is usually associated with greater fatigue. When this variable decreases, the muscle depends more on the anaerobic pathway, generating a greater accumulation of waste products such as lactate [36]. To reduce these metabolites, rest time in strength training must be adjusted to allow optimal reoxygenation between sets.